# In Vivo Sensing of pH in Tomato Plants Using a Low-Cost and Open-Source Device for Precision Agriculture [note 1]

**DOI:** 10.3390/bios12070447

**Published:** 2022-06-23

**Authors:** Antonio Ruiz-Gonzalez, Harriet Kempson, Jim Haseloff

**Affiliations:** Department of Plant Sciences, University of Cambridge, Downing St., Cambridge CB2 3EA, UK; a.gonzalez.16@ucl.ac.uk (A.R.-G.); hljk2@cantab.ac.uk (H.K.)

**Keywords:** pH sensor, precision agriculture, metal oxide

## Abstract

The development of sensing devices for precision agriculture is crucial to boost crop yields and limit shortages in food productions due to the growing population. However, current approaches cannot provide direct information about the physiological status of the plants, reducing sensing accuracy. The development of implanted devices for plant monitoring represents a step forward in this field, enabling the direct assessment of key biomarkers in plants. However, available devices are expensive and cannot be used for long-term applications. The current work presents the application of ruthenium oxide-based nanofilms for the in vivo monitoring of pH in plants. The sensors were manufactured using the low-cost electrodeposition of RuO_2_ films, and the final device could be successfully incorporated for the monitoring of xylem sap pH for at least 10 h. RuO_2_ nanoparticles were chosen as the sensing material due to its biocompatibility and chemical stability. To reduce the noise rates and drift of the sensors, a protective layer consisting of a cellulose/PDMS hybrid material was deposited by an aerosol method (>GBP 50), involving off-the-shelf devices, leading to a good control of film thickness. Nanometrically thin films with a thickness of 80 nm and roughness below 3 nm were fabricated. This film led to a seven-fold decrease in drift while preserving the selectivity of the sensors towards H^+^ ions. The sensing devices were tested in vivo by implantation inside a tomato plant. Environmental parameters such as humidity and temperature were additionally monitored using a low-cost Wio Terminal device, and the data were sent wirelessly to an online server. The interactions between plant tissues and metal oxide-based sensors were finally studied, evidencing the formation of a lignified layer between the sensing film and xylem. Thus, this work reports for the first time a low-cost electrochemical sensor that can be used for the continuous monitoring of pH in xylem sap. This device can be easily modified to improve the long-term performance when implanted inside plant tissues, representing a step forward in the development of precision agriculture technologies.

## 1. Introduction

The development of precision agricultural methods has become essential to tackle current challenges in food security. This technology has the potential to tackle the 70% increase in agricultural production needed to mitigate the demands of the growing population by 2050 [1,2]. To achieve high-quality monitoring, precision agriculture requires the use of sensors that can provide information about crop health [3,4], ideally at an individual level [2]. However, the field of implantable sensors in crops has been dominated by spectral imaging techniques, often requiring the pre-implantation of nanomaterials such as carbon nanotubes, and determining the changes in fluorescence [5]. However, in most cases, these sensors are limited to the detection of simple reactive oxygen species (ROS), such as H_2_O_2_, or nitroaromatics [6], and require expensive optical equipment for the imaging of the implanted nanoparticles, hindering their application within real-world environments.

Electrochemical technologies for the monitoring of internal analytes in plants have proven to be promising alternatives for the low-cost fabrication of implanted sensors that can continuously monitor biomarkers. These methods could be used for a wide range of analytes and applications, including the monitoring of hormone fluxes in roots through the use of carbon nanotube-based self-referenced microelectrodes [7]. Recently, the monitoring of the relative concentrations of certain leaf biomarkers such as glucose and fructose has been made possible through the incorporation of transistor-based devices [8]. Such a sensor showed for the first time an increase in fructose concentrations during the night-time. However, the determination of single ions such as H^+^ still represents a challenge in the field of implanted sensors in plants due to the scarcity of suitable materials for their determination.

The direct determination of pH inside plant stems represents a promising approach to tackle the early diagnosis of plant diseases. An increase in plasma cell acidity has been found to change as a response to Fusarium Oxysporum infections in Arabidopsis [9]. In the case of tomato plants, xylem pH has been suggested to change due to transpiration [10], nitrogen sources [11], and a small increase has been measured in night-time compared to daytime extracts [12]. As such, xylem pH represents a powerful multimodal biomarker that could be used for the monitoring of plant physiological status. By combining the information from in vivo pH and environmental conditions, which can be tackled using low-cost sensors for temperature, humidity, and other factors, an accurate diagnosis of plant diseases can be achieved.

The use of metal oxide-based sensors for the determination of pH has attracted great interest within the past few years, given their good sensitivity, combined with fast response and long lifetime [13]. Specifically, the use of iridium and ruthenium oxide materials has risen interest within the past few years since they can selectively react with H+ ions, leading to a near-Nernstian potentiometric response [14]. Iridium oxide represents one of the most used pH-sensing materials used nowadays [13], showing a good biocompatibility [15], and being applicable for in vivo studies. However, a significant change in response has been recorded in this material due to the presence of dissolved oxygen [16,17], which can enhance the drifts and lead to a systematic deviation from Nernstian behaviour. On the contrary, ruthenium oxide-based sensors have been proven to have a lower response towards dissolved oxygen [18] and can be deposited using low-cost methods such as electrodeposition [19]. Consequently, these materials have been widely explored for miniaturisation purposes within Organ-on-a-Chip devices [18], or even implanted sensors in human patients [20]. However, these sensors lack good stability and repeatability due to the variability in the sensing response of the metal oxide nanoparticles driven by the shape, surface area and dimensions [13]. In addition, the equipment required for the fabrication, characterisation and testing within real-world applications tends to be relatively expensive, limiting the applicability within low-resource settings. As such, new approaches for the development of low-cost and stable pH sensing devices are needed, to enable the monitoring of plant health for smart agricultural applications.

In the present work, a pH sensor based on electrodeposited ruthenium oxide nanoparticles onto copper electrodes and containing a protective cellulose:PDMS film has been fabricated. This electrode could be implanted inside tomato plants for the continuous monitoring of pH. Thin RuO_x_ films were initially fabricated by a custom-made potentiostat device, involving a microcontroller and two operational amplifiers (>GBP 50). The fabrication results were compared to the ones obtained using a laboratory-standard equipment (Metrohm, Autolab). To improve the stability of the films and enable the monitoring of xylem pH in vivo, a cellulose:elastomer film was additionally incorporated onto the RuO_x_ electrodes. This layer was crucial to improve the stability of the sensing films both in terms of drift and noise rates. Such an increase in stability was key to enable a long-term monitoring of plant pH. Finally, the devices were tested implanted inside a 4-week-old tomato plant, and the results, along with the temperature, humidity and volatile organic compounds (VOC) data collected by an environmental sensor, were wirelessly reported to an online server to simplify the data collection process. The biological response of plant tissues was also characterized, both in terms of the changes in tissue morphology and the impact on the sensor properties, which could be used for the future design of implantable sensors for long-term monitoring of xylem sap biomarkers. As such, this work represents a step forward in the monitoring of chemical biomarkers in situ using low-cost devices, with potential applications in food security and disease prevention in plants, among others.

## 2. Materials and Methods

### 2.1. Materials

Reagents were purchased from Sigma Aldrich, unless otherwise specified: ruthenium (III) chloride, ethyl cellulose, conductive silver ink, and iron (III) chloride, ethanol, Toluidine Blue O. All pH buffer solutions, glass slides, and glass cover slips were purchased from Fisher scientific Ltd., UK. Sylgard 184 silicone elastomer kit was purchased from Dow. A Wio Terminal microcontroller was purchased from Seeed Studio, MCP4725 and BME680 were purchased from Pimoroni. Arduino pH-4502C pH meter was purchased from Morden Store. Finally, LM324 Operational amplifier was purchased from Texas Instruments. Heating element (4in, 400 W, 220 V ac) and temperature sensor (TMP36) were purchased from RS components. Double edged razor blades were purchased from Wilkinson Sword.

### 2.2. Assembly of Low-Cost Potentiostat

The pH sensing electrodes were fabricated by electrodeposition using a custom-made and low-cost potentiostat. This potentiostat was assembled by following reported circuits within the literature [21,22,23]. Briefly, an LM324 operational amplifier was used combined with an MCP4725 Digital-to-Analog converter (DAC), that allowed the user to set a specific voltage output. This MCP4725 was connected to the non-inversing input of an operational amplifier, while the counter and reference electrodes were connected to output and inverting electrodes, respectively. On a different operational amplifier, the working electrode was connected to the inverting input, and a current-to-voltage converter circuit was designed by a closed-loop configuration, using a 15 kOhm resistor between the inverting input and the output. This circuit was connected to the analogue pin of a Wio Terminal microcontroller, which could be used to set the potentiostat parameters by the user. This circuit has been schematised on Figure A1.

### 2.3. Electrodeposition and Characterisation of Ruthenium Oxide Films

Within the present work, a thin RuO_x_ film was used for the pH sensing. Thin RuO_x_ films were fabricated onto copper electrodes by using electrodeposition as reported elsewhere [24]. Briefly, copper electrodes were immersed onto a solution containing 0.1 M RuCl_3_, and an increasing voltage was applied. This electrodeposition process was conducted initially using a benchmark laboratory equipment (Metrohm, Autolab BV, The Netherlands) employing a Ag/AgCl reference and a platinum film as counter electrodes. A voltage comprised of between 0–0.8 V was applied at a rate of 10 mV/s, and the current was measured to make a Cyclic voltammogram. The cyclic voltammetry process was repeated 10 times to develop the metal oxide films. This deposition method was replicated using our custom-made potentiostat, applying the same conditions in the case of the benchmark laboratory equipment, and the analogue response recorded by the microcontroller was recorded.

After the fabrication of RuOx, the films were visualised by scanning electron microscopy (EVO LS15, ZEISS, Jena, Germany), using an acceleration voltage of 20 kV. This method allowed the characterisation of the surface structure of the sensing films. The morphology of RuOx films fabricated by standardised laboratory equipment and our custom-made potentiostat under abovementioned conditions were then compared. The chemical structure of the RuO_x_ sensing film was additionally studied by using FTIR (L160000A Perkin Elmer, Waltham, MA, USA). This method allowed the determination of the presence of the stretches corresponding to the Ru-O bonds, and presence of surface hydroxyl groups that are necessary for the sensing. In this case, the transmittance of the deposited films was measured between 400–4000 cm^−1^ using an ATR configuration. Finally, the sensing performance of the developed devices in terms of sensitivity and electrochemical noise was determined by calibrating the electrodes using multiple pH buffers ranging from 4–11. In this case, a Ag/AgCl reference electrode was designed by coating a copper electrode with silver paint. This silver paint was further oxidised by leaving the electrodes in contact with a 0.1 M solution of FeCl_3_ for 1 min.

### 2.4. Incorporation of Cellulose-Based Thin Films

To improve the performance of the sensing films, a cellulose: PDMS film was deposited onto the RuO_x_ film. This deposition was carried out by using a custom-made aerosol deposition method as reported previously [25]. The deposition system consisted of an ultrasonic atomiser operating at a frequency of 110 kHz, and a low-cost air pump. The sensing films deposited onto the copper electrodes were placed onto an aluminium substrate that had been heated up to 40 °C. This temperature on the substrate was achieved by combining a heating element and a temperature sensor controlled by an Arduino device. This Arduino was plugged to a relay which could be used to set the desired temperature on the substrate.

A solution containing 40 mg ethyl cellulose cellulose and 60 mg PDMS diluted in 10 mL ethyl acetate was then aerolised and directed towards the desired substrate as an aerosol. Initially, the thickness obtained by this method was characterised by fabricating films using different deposition times. To do so, the films were deposited onto glass substrates, and the obtained thicknesses were determined by using a stylus profilometer (Dektakxt, Bruker, UK). The arithmetic average roughness of the films was additionally assessed to study the film homogeneity after the deposition.

The final devices, containing the RuO_x_ films fabricated by our custom-made potentiostat onto the copper electrodes, were modified by the deposition of 80 nm thin films (Figure 1a). The sensitivity of these sensors was characterised by subjecting the electrodes to different pH buffers, and the voltage was monitored. The changes in the electrochemical drift and selectivity of the devices were additionally determined. The selectivity coefficients of both pristine electrodes, where only the RuO_x_ films had been deposited, and cellulose-coated sensors were measured following the Matched Potential Method. Moreover, the drift was determined in both cases by immersing the electrodes in a solution with a pH of 8 and continuously recording the changes in the potential of the electrodes for 1 h. After the sensors were characterised, the RuO_2_- modified electrodes and solid-state Ag/AgCl reference were implanted inside a 4-week old tomato plant to enable a monitoring of xylem pH (Figure 1b).

### 2.5. In Vivo Testing of Electrodes

To test the feasibility of the devices within an in vivo environment, the sensing devices were implanted inside 4-week-old tomato plants (Moneymaker variety). Sensing devices containing the electrodeposited RuO_x_ sensing films and Ag/AgCl reference electrodes were fabricated using the low-cost potentiostat reported here. Initially, electrodes where no cellulose-based film had been incorporated were tested, and the voltage was monitored for 5 h. This experiment was conducted using a Wio Terminal device as the microcontroller, enabling the wireless reporting of results to Adafruit IO server. Devices containing both the RuO_x_ and the cellulose-based coating were also produced and tested. In this case, the sensors were also implanted in 4-week-old tomato plants, and the voltage was monitored using a Wio Terminal device. In addition, a BME680 environmental sensor was incorporated to prove additional information about the environmental conditions of the plant.

### 2.6. Characterisation of Biological Response to Sensing Electrodes

The changes in the electrical properties on the surface of the electrodes could initially be quantified by measuring the electrical impedance of electrodes, in this case, a variable voltage of 0.1 V amplitude was applied between the reference and working electrodes within a range of 10^−1^–10^5^ Hz. The total magnitude of the impedance was compared. This electrical impedance was measured in both the pristine and cellulose-coated electrodes on day 1 and day 7 of implantation.

Finally, the changes in signal drift of the electrodes were determined. In this case, the signal obtained from the sensing electrodes was recorded for 1 h after the implantation inside the tomato plant. After 7 days of continuous implantation, the electrochemical signal was continuously recorded again, allowing a comparison between the intial drift in plant sap, and the increase due to tissue responses.

### 2.7. Staining and Imaging Tomato Stem Cross Sections

For the staining and imaging, a protocol adapted from the work of Mitra et al. [26] was used. Free hand sections of tomato stems made using a wet (Wilkinson Sword) razor blade. The sections were stored in reversed osmosis (RO) water until all made. They were submerged using an inoculation loop or forceps in 0.02% Toluidine Blue O solution for a duration of between 30 and 60 s. Samples were carefully transferred 15% ethanol solution to develop colour differentiation for one minute. They were then stored once again in pure water. Slides were prepared by dropping RO water onto a glass slide using a Pasteur pipette. One sample was mounted in the water drop and a 22 × 22 millimetre square glass cover slide coverslip was carefully placed over, avoiding air bubbles. These prepared slides were viewed and imaged using a Keyence digital microscope, magnification ranging from 20× to 200×. Scale bars were created using the Keyence operating software.

## 3. Results and Discussion

### 3.1. Deposition and Characterisation of pH Sensing Films

One of the key steps in the fabrication of the pH sensors is the deposition of a sensitive film. Within this work, ruthenium oxide was employed as the active layer, given its ability to reversibly reduce the surface RuO_2_ using H^+^ groups according to the formula below [27]:RuO_2_ ∙ H_2_O + H^+^ + e^−^ → H_2_O + Ru(OH)_3_

This material shows a better stability compared to similar pH-sensing compounds such as IrO_2_ [16,17,18]. In this case, the RuO_2_ sensing films were deposited by electrodeposition, using a copper substrate as the working electrode, a Ag/AgCl reference and a carbon-based counter. In this case, a voltage between 0 and 0.8 V was applied at a rate of 100 mV/s, and the resulting current was determined. Up to 10 cycles were applied, and the results using standardised laboratory equipment (AUTOLAB, Metrohm) and a low-cost potentiostat device reported here were compared.

After the fabrication of RuO_x_ films through electrodeposition directly onto the copper electrodes using laboratory-standard equipment and our low-cost microcontroller-based solution, the electrodes were characterised. Initially, the chemical composition was studied using FTIR, which could be used for the determination of chemical stretches on the nanoparticles. The presence of hydroxyl groups and Ru-O bonds on the RuO_x_ nanoparticles is crucial, since they are responsible for the H^+^ sensitivity as previously shown [27].

The FTIR revealed the presence of the Ru-OH stretches at 821 cm^−1^ [28] in both cases. In addition, the stretch corresponding to the peroxo groups was observed at 1014 cm^−1^ [29], along with the H_2_O stretches at 1600 cm^−1^ [30] and the ones from hydroxyl groups at 3400 cm^−1^ [31]. These groups reflected the adsorption of water molecules onto the RuO_x_ nanoparticles, which is key for H^+^ sensing. These stretches were present within the samples fabricated using both Autolab- and Arduino-based equipment, proving the suitability of the low-cost method for the fabrication of pH sensing films (Figure 2a).

The morphology of the final sensing films was additionally evaluated by using SEM. Initially, pristine copper electrodes prior the deposition of films were studied. In this case, a homogeneous surface was obtained, with no presence of nanoparticles (Figure 2b). After electrodeposition, a film of RuO_x_ nanoparticles was observed on the copper electrodes, with a size in the range of 200 nm, showing the deposition of the sensing material (Figure 2c). Similar structures have been observed for the determination of pH changes in Lab-on-a-Chip devices [18] and water quality monitoring [13]. A similar morphology to the one obtained using the Autolab equipment was observed in the case of low-cost electrodeposition with the Arduino-based potentiostat, showing the presence of nanoparticle film onto the surface of the copper-based electrodes (Figure 2d).

Our Arduino-based approach led to the successful deposition of sensing films onto the copper substrates with similar morphologies and grain sizes. These nanoparticles also showed a similar chemical composition as determined by FTIR. These films could also be used to determine the pH of solutions using a low-cost device as determined within the following sections.

### 3.2. Electrochemical Characterisation of Sensing Films

To test the initial sensitivity of the films after deposition using the lab-proof potentiostat, a two-electrode cell was prepared by coating a copper electrode using a silver ink. This silver ink was used as a Ag/AgCl electrode after oxidising it by using FeCl_3_ for 1 min following previous work in the field [32]. The full device was immersed in different pH buffers, and the voltage of the cells was recorded (Figure 3c,d). In the case of the low-cost Arduino device calibration, an off-the-shelf device was employed for the voltage measurements.

When the sensing devices were characterised using laboratory equipment, a Nernst sensitivity of 53.7 mV/pH was achieved. This value demonstrated the suitability of our potentiostat in the fabrication of pH-sensitive electrodes, since it was similar to the Nernst sensitivity of 60 mV/pH sensitivity, which is the current standard for ion-selective electrodes. When the Arduino-based device was employed instead, a good linearity was observed within the measured pH range (R^2^ = 0.987), similar to the results obtained by the laboratory equipment, with a sensitivity of 30.4 points/pH. This performance allowed an accurate determination of the pH of solutions within biological environments, since the chosen range comprises the pH values typically observed within tomato plant xylem saps [12].

The low-cost approach described within the present work for the fabrication of the sensing films comprised a Wio Terminal device, which allowed the control of the components involved, and an interactive interface for the selection of the desired voltage range and speed by the user. This system could be complemented with multiple environmental sensors of interest such as the BME680 for temperature, humidity and gas measurements, and could additionally analyse the results in real-time. The whole circuit was enclosed within a plastic housing, and the three electrode cells were connected to the potentiostat (Figure 3e,f). A schematical representation of the device circuit is shown in Figure A1. In addition, the necessary codes have been made available on XOD.io, for object-oriented programming (Figure A2).

Our results show that microcontroller-based potentiostats can be used for the deposition and pH sensing of RuO_x_ nanoparticles, with similar sensitivities when compared to laboratory equipment. This microcontroller device greatly reduced the production and operation costs and enabled the portability of the developed devices, showing promise for their real implementation within real-world settings. The RuO_x_ films showed a Nernst response when used in the presence of different pH buffers, and a good linearity was observed within the measured pH range (4–9). This level performance is advantageous for the development of implanted devices for plant monitoring. However, one of the limitations of the devices was the high noise rates that were seen, in the range of 0.16 mV/min, and 0.24 Point/min in the case of the Arduino-based devices. These noise rates could hinder applications in biologically relevant environments. As such, further modifications using cellulose-based coatings were tested to improve the performance of the sensors.

### 3.3. Deposition of Cellulose-Based Film on Sensing Device

The deposition using the Autolab equipment and microcontroller-based devices led to similar morphologies and sensing performances, showing promise for the incorporation of these low-cost methods in the fabrication of implantable devices. However, the noise rates of the films were high, which could lead to large uncertainties on the measured pH, especially given the logarithmic nature of this value [33].

To improve the performance of the sensing films by reducing the electrochemical noise, a cellulose-based coating containing PDMS elastomer at a ratio of 60:40 was developed. This coating composition was selected following previous reported work that indicated an improvement in sensing stability of ion-selective electrodes using elastomer coatings [34] and a potential decrease in biomolecule adsorption when PDMS is combined with ethyl cellulose [25]. As such, it could be applied to improve the stability of the pH-sensing films, which could monitor biomarkers in biological solutions.

Thin films were deposited onto the RuO_x_ films using an aerosol-based method (Figure 4a). The aerosol method developed in this work consisted of a piezoelectric atomizer, which could operate at a frequency of 110 kHz, and a low-cost air pump that could be connected to a microcontroller (Figure 4a). The use of an air pump allowed the displacement of the generated aerosol from the precursor solution reservoir onto the electrodes, generating a uniform film with a roughness in the range of 3 nm. Initially, this deposition method was applied to copper wires to compare the performance with the drop casting method (Figure 4b,c). A higher surface roughness was observed in the case of drop cast samples compared to the aerosol deposited films, evidencing a higher surface homogeneity of the case of aerosol deposited films. This improved homogeneity was key to produce sensing films with reproducible performances and reduce the adsorption of biomolecules. In addition, the thickness of the films fabricated by aerosol deposition could be controlled by changing the deposition time, which allowed a good control of the dimensions, with an estimated growth of 0.77 nm/s (Figure 4d).

The presence of a cellulose-based coating could reduce the measured electrochemical noise, from 0.24 ± 0.02 Point/min, obtained in the case of uncoated electrodes, up to 0.19 ± 0.01 Point/min (Figure 4e). In addition, the electrodes showed a similar sensitivity compared to the pristine RuO_x_ electrodes, with 35 point/pH, and a linear response within the same pH range as the uncoated electrodes (Figure 4f).

To allow a good control over the film dimensions and morphology, a heating element was incorporated within our low-cost deposition method. This heating element allowed us to obtain a homogeneous temperature for the evaporation of the carrying solvent. All film precursors had previously been dissolved in this carrying solvent, allowing the deposition of the films onto the copper conductive electrode. Deposition temperature is crucial during aerosol deposition, since it can greatly influence the roughness of the deposited films, thereby having an impact on the sensing performance [25]. In this case, the temperature of the substrate was controlled using an analogue temperature sensor physically attached to the aluminium substrate where the sensing device was placed. The temperature readings from this sensor were used to regulate the substrate temperature via an Arduino microcontroller connected to a relay to power the heating element. By using this system, stable temperatures with a small deviation of ±1 °C could be obtained (Figure 5a). To achieve a low roughness of the films, a moderate temperature of 40 °C was applied enabling the evaporation of ethyl cellulose. The final deposition method led to a low film roughness of approximately 3 nm.

Although the application of a relatively high temperature was crucial to achieve a high homogeneity and good thickness control, in some cases, it can also promote the degradation of the film components. As such, to study the stability of the film components after deposition, a pristine PDMS film and an ethyl cellulose film were deposited onto two different glass substrates each at a temperature of 40 °C. FTIR was then conducted on both films, and the presence of characteristic stretches from both components was analysed (Figure 5b,c). In the case of PDMS, the stretches at 1260 cm^−1^, 2960 cm^−1^, 1076 cm^−1^, and 796 cm^−1^ were observed, as expected from the structure of PDMS [23]. These peaks correspond to the CH_3_ symmetric bending, C-H stretching, Si-O-Si bond, and CH_3_ rocking, respectively. The presence of these peaks was indicative of a non-thermal degradation of PDMS during the deposition [35]. After the study of PDMS, the stability of ethyl cellulose films deposited by aerosol was studied. Cellulose derivatives display multiple stabilities due to the presence of different functional groups over their surfaces. The observed FTIR spectrum in this case was consistent with the reported work, with stretches located at 1054 cm^−1^, 1380 cm^−1^, and 3499 cm^−1^ consistent with the presence of C-O, CH_2_ and -OH stretches, respectively [36]. The obtained FTIR spectra from both components was indicative of no degradation of the protective film components upon deposition using the aerosol method here described. This stability was key for in vivo testing of the final devices and had an impact on the electrochemical performance of the sensors.

As previously observed, the incorporation of this protective film led to an increase in signal stability of the sensing films evidenced by the lower noise rates. To further demonstrate the increase in performance on sensing when using protective films, the electrochemical drift was assessed (Figure 5d). This drift was measured by subjecting the pristine and coated electrodes to a solution with a pH of 8 and recording the changes in the measurements over 1 h. The results indicate a 7-fold decrease in drift, changing from 18.1 ± 3.1 to 2.5 ± 1.3 mV h^−1^.

As observed, the modification of sensing devices using a cellulose-based coating led to an improvement in the stability of pH sensors. This stability was reflected in lower noise and drift rates, showing promise for the incorporation in biological tissues. A key aspect of the sensing performance is the selectivity of devices due to the high presence of electrolytes inside plant tissues. In the case of tomato plants, potassium could be considered one of the most important interferences, given its concentration of approximately 40 mM, as measured by Gallegos-Cedillo et al. [12]. As such, the selectivity of the sensing devices was evaluated. In this case, the matched potential method was used, and selectivity coefficients of $k_{pH,\;K^{+}}^{MPM}=-3.1$ and $k_{pH,\;K^{+}}^{MPM}=-3.2$ were obtained for the pristine and cellulose-modified electrodes, respectively. These selectivity values were superior when compared to the coefficients obtained on similar metal oxide-based electrodes, such as iridium oxide thin films, which had values of approximately −2.24, as reported by Huang et al. [37]. In addition, the similarities between selectivity coefficients in both cases indicate a non-interference of the protective coating with the sensing process carried out by the RuO_x_ nanoparticles.

As demonstrated, the incorporation of this cellulose-based coating onto the electrode surface improved the noise rates of the sensors when compared to the pristine metal oxide-based device. This coating could also improve the long-term stability of the sensors which proved crucial when incorporated during in vivo studies, as shown in the next section. These improvements in the drift and noise rates were likely a consequence of the lowest adsorption of biomolecules, as described in previous reported work [25]. The use of ruthenium oxide also led to a higher selectivity when compared to other metal oxide-based sensors, representing a promising approach for the development of low-cost devices for the in vivo monitoring of pH.

### 3.4. Testing of Final Device for In Vivo Plant pH Monitoring

After the fabrication and optimisation of the sensing devices through the incorporation of a cellulose-based coating, the electrodes were implanted inside tomato plants. The working and reference electrodes (1 mm wide) were glued onto a PP substrate to facilitate the implantation (Figure 6a,b), and they were directly inserted onto the tomato stem (Figure 6c). A Wio Terminal-based device was assembled, incorporating a low-cost voltage metre, and a BME680 sensor, able to measure environmental humidity, temperature, pressure and VOC concentrations. The device was also programmed to send data wirelessly to an online server for data analysis and could be directly powered using an inductive charger (Figure 6d,e). All the necessary components for the environmental measurements were enclosed inside a portable box that enabled the easy handling of the devices. Finally, the results were downloaded from an online server, enabling the wireless reporting of results.

After the assembly of the low-cost system, the electrodes were implanted inside tomato plants. Initially, electrodes where no cellulose coating had been deposited were tested. The initial baseline was first recorded for 1 h, stabilizing at a pH of 2.42 ± 0.84. The plant was then watered using 20 mL DI water, increasing the pH to 3.44 ± 1.52 (Figure 7a). However, given the increasing noise rates obtained in this case, an objective study of the plant response became difficult. Such high electrochemical noise was attributed to a protein adsorption and biofouling, which commonly takes place within implantable devices that incorporate similar materials such as iridium oxide [38].

When the RuO_x_ sensing film was coated using a cellulose/elastomer film, the noise rates of the measured signal improved significantly compared with the pristine pH sensors. A pH baseline of 4.17 ± 0.03 was initially recorded when the devices were first implanted. This value was similar to previous measurements of xylem sap pH in tomatoes, which reported a slightly acidic value [10]. Upon watering the plant, an initial increase in pH was observed, followed by a stabilisation of the value at 4.37 ± 0.07 (Figure 7b). Changes in pH values have been observed in vascular plants due to an increase in transpiration [10], change of season [39], presence of light [40] or fertilisation using nitrates [41]. As such, the increase in pH was attributed to a higher water availability by the plant, demonstrating the potential application of the device for the continuous monitoring of pH in vivo. This low-cost system could also potentially be employed in smart agriculture, enabling the study of plant infections [42] and wound response [43], which could save costs and enhance productivity.

To complement the information obtained by the in vivo devices, we incorporated a BME680 environmental sensor, which recorded information about the temperature, humidity and VOCs concentrations (Figure 7c–e). The built-in light sensor from the Wio Terminal Device was additionally used to study the day-night cycles of plants (Figure 7f). As expected, the magnitude of environmental humidity increased upon watering the tomato plant. As such, this sensor, combined with the pH information from the implantable device, could offer information about the water or nutritional needs of the plant. The resistance measured by the VOC sensor also increased. However, in this case, given the nature of the resistor-based VOC sensor, this increase in resistance can be translated into a lower concentration of VOCs. The information recorded form other physical sensors such as light and temperature was correlated, with lower temperatures recorded at night-time. These sensors could be exploited in the future for adjusting the growing conditions of plants when they are kept under controlled environments.

As expected, the decrease in daylight was linked to a decrease in environmental temperature and an increase in humidity. In addition, the pH inside the stem increased after the stabilization due to watering increased, with a value of 4.37 ± 0.07 compared to the original baseline. This slight increase in the pH values of tomatoes during night-time has been previously described in tomato plants. This increase has been shown to be a consequence of differences in transpiration between day and night rather than changes in electrolyte concentrations in xylem [25].

### 3.5. Materials Biocompatibility for Long-Term Applications

One of the major challenges in the field of implanted biosensors is tissue compatibility with the host organism. Due to the healing process taking place at the interface between tissue cells and sensors, the accuracy of measured pH is decreased over time. In the case of implanted devices in humans, the foreign body response has been widely documented, involving the inflammation of surrounding tissue [44] and the formation of a fibrotic tissue that isolates the sensor surface [45]. However, this process is not well-understood in plants, limiting the development of sensing approaches in the field. As such, we conducted a study to determine the impact of the pH sensors developed in this work on the tissue morphology and healing process, as well as the changes due to the modification of devices using ethyl cellulose.

Both pristine sensing devices, containing the RuO_x_ films alone, and the modified sensors containing ethyl cellulose were initially implanted onto the tomato plants. A cross-section of the stem where the sensors had been implanted was imaged by optical microscopy in the same day that the sensors had been implanted (Figure 8a,b). In this case, a Toluidine Blue O-based stain was used. This cationic dye binds to cell wall components such as lignin or pectins [26], allowing a visualization and identification of the cells contained in the pith and xylem due to the differences in composition. Plant xylem is a vascular tissue responsible for the transport of water and electrolytes, including electrolytes such as potassium ions and nitrate [25]. As such, we targeted xylem for the study of pH variations in plants. This tissue was located within 600 µm from the tomato plant stem epidermis and showed a thickness of 500 µm, as measured from optical micrographs.

Within the first hour after the insertion of the electrodes inside the plant xylem, no significant wound response could be observed from the plant. As determined in the previous section, in both cases, the electrodes were put into contact with the sap and could determine the concentration of xylem sap pH for at least 1 h. Due to the continuous contact with the xylem sap solution, the magnitude of the electrical impedance was low in pristine and cellulose-modified electrodes, showing an average value of approximately 3 kΩ in both cases (Figure 8c). The impedance recorded within the lowest frequency range, being indicative of the electrical resistance of the films, was 22.01 and 22.99 kOhm for the uncoated and EC/PDMS protected RuO_x_ films, respectively. These results indicate a relatively low electrical resistance of the nanoparticulate coatings and a low resistance of the thin EC/PDMS films. As such, the incorporation of the protective coating film showed a low electrical interference in the electrical properties of the electrodes. In the case of the high frequency range of the impedance plots, resistance values of 76.97 and 61.31 Ohms were measured, indicating a low ionic resistance, which is crucial for the sensing of H^+^ ions. The values of real and imaginary impedance recorded at different frequencies could additionally be used to estimate the changes in tissue resistance by fitting the results into an equivalent circuit (Figure A3). In this case, a resistor and capacitor in parallel were used for the modelling, as reported previously [46]. This modelling could be used to measure both the intracellular and extracellular resistance as well as the cell capacitance, which is indicative of the movement of ions within the cells. In this case, the extracellular and intracellular resistance were similar, at approximately 8 kOhm and 80 Ohms, respectively. In addition, the capacitance was approximately 10 µF.

However, after 7 days of continuous contact with the xylem tissue, this impedance increased to 90.4 kΩ for the pristine electrodes and 72.5 kΩ in the case of the cellulose-coated sensors. This increase in the electrical impedance of the sensors over time was attributed to the formation of necrotic tissues surrounding the electrodes, with a high content of lignin. Similar responses from plants have been observed during the grafting process, whereby a defence response is triggered to seal the tissues from the plants [47]. In this case, the obtained resistance within low-frequency ranges was approximately 947 and 811 kOhm in the case of pristine and EC/PDMS-coated sensors, respectively. The lower resistance obtained in the case of the EC/PDMS was indicative of a lower lignification of tissues around the sensor area, suggesting a better biocompatibility of the coating. On the contrary, the resistance obtained within the high-frequency range was similar to the one obtained in the previous case on day 1, being 58.51 and 64.20 Ohms. When the impedance plots were simulated using the same equivalent circuit, the intracellular electrical resistance was similar to the one obtained in the previous case, being approximately 60 Ohms. On the contrary, the extracellular resistance greatly increased, reaching 1.84 MOhm in the case of uncoated sensors and 1.16 MOhms when EC/PDMS-coated devices were employed. This increase in extracellular resistance was consistent with the formation of the necrotic tissue area around the sensing electrodes. The calculated capacitance also decreased, from 10 µF measured on day 1 to 1 µF after a week of continuous exposure to the plant tissue. As a consequence of this wound-healing process, the contact of sap solution with the sensor surface is restricted, reducing the accuracy of the sensing devices. Due to this healing process, the drift of the sensors also increased (Figure 8d).

These results suggested the formation of a necrotic tissue film surrounding the pH sensing electrodes, leading to an increase in impedance and electrical drift. This drift increased by over 10-fold after 7 days, changing from 6.8 mV h^−1^ to 71.5 mV h^−1^. These measurements were taken after re-equilibrating the pH sensors for 7 days. In this case, the devices were kept implanted inside the tomato plants, but no electrochemical measurements were taken between day 1 and day 7. As such, this high value in drift was influenced by the initial drift from initializing the OCP sensing device, and the wound healing mechanism of the plant, which restricts the sap flow onto the sensing area. In addition, it is expected that the wound healing mechanism in the plant involves the production of metabolites and the release of electrolytes which might compromise the composition of the sensing films within the sensing area. These phenomena explain the relatively similar pH recorded in both day 1 and day 7 despite the high drift of the devices.

As mentioned, the healing process leads to a decrease in sensing accuracy, reflected in a high electrochemical drift. The lignification of the tissues around the sensing electrodes could be observed after 7 days of continuous contact of the electrodes with the films. The formation of necrotic tissue was visible even without the staining using Toluidine Blue (Figure 9a,b). Upon examination with the microscope, a dark tissue around the edges of the wound, not observed in day 1 microscopy pictures, was present. Non-disorganized cells of variable sizes were present around this dark tissue layer, being indicative of the regenerative process of the tissues (Figure 9c,d).

When cellulose-coated sensors were implanted in the tomato plant, a similar response was observed. However, the thickness of the necrotic tissue surrounding the sensor was significantly lower than the one in the case of pristine electrodes. In addition, the cells surrounding this lignified tissue did not share the disorganized morphology obtained with pristine electrodes, indicating a better integration of the sensors with the plant. The lower thickness of the lignified tissue around the electrodes was also consistent with the lower impedance magnitude measured.

The study of biological responses from plant tissues towards the electrodes and cellulose-coated electrodes showed a limitation of the incorporation of implanted sensors in long-term applications. Although this process could not be prevented by the incorporation of the protective film, the plants showed a lower degree of tissue damage, and the obtained drift was significantly lower than that of the pristine RuO_x_ films. Thus, this work demonstrates the need for the development of materials that can minimize the tissue responses and enable a true monitoring of biomarkers in plants.

## 4. Conclusions

In the present work, a low-cost miniaturised device based on ruthenium oxide coated with a cellulose-based film was developed to enable the in vivo monitoring of sap pH in tomato plants. Initially, a RuOx film was fabricated onto copper electrodes by electrodeposition. An Autolab potentiostat/galvanostat was used as a benchmark equipment for the fabrication, and the results were compared to those obtained by using a microcontroller-based low-cost approach. Nanoparticulate films were obtained in both cases, with a similar morphology and chemical compositions that allowed the sensing of pH with a Nernstian slope. However, the electrochemical noise rates of the devices were relatively high, in the range of 0.24 point/min, hindering the applicability of this system for the direct determination in vivo.

To improve the stability of the sensing devices and allow the implantation of the films, a cellulose-based coating was incorporated onto the electrodes. This film was deposited by using an aerosol-based method, which included an ultrasonic sonicator and an air pump. A heating system was additionally incorporated to improve the film homogeneity after the deposition. The final low-cost system (>USD 30) allowed the deposition of nanometrically thin films with controllable dimensions on multiple substrates. The stability of the deposited components was additionally assessed by FTIR, evidencing the non-degradation of film components after the deposition. A growth of 0.77 nm/s was measured by a stylus profilometer, with a roughness below 3 nm. Films within a thickness of 80 nm were then deposited, and the final device was calibrated using different pH buffers. A significative reduction in the noise rates was obtained, in the range of 0.19 point/min, while the measured sensitivity resulted similar to the one measured by the pristine electrodes, with no cellulose-based coating.

The increase in stability of the sensors due to the incorporation of a cellulose-based coating was additionally reflected by a decrease in the drift rates of the sensors, which changed from 18.1 mV h^−1^ to 2.5 mV h^−1^. This improvement in electrochemical drift proved beneficial during the in vivo testing inside plant tissues. Moreover, the presence of this cellulose-based coating did not show a significant impact on the selectivity of the devices towards H^+^, especially compared to K^+^, one of the main interfering ions found in xylem sap.

Finally, the devices were implanted inside tomato stems, and the pH was continuously monitored for 5 h. When no coating was incorporated, an increasing electrochemical noise was measured, which did not allow the determination of plant pH. In contrast, the final device incorporating the cellulose-based coating could monitor the pH in the sap continuously for at least 5 h, with a low noise in the range of 0.07 pH/min. Initially, a stable pH of 4.17 ± 0.03 was measured. Upon watering the plant using DI water, a sudden increase in this value followed by a stabilisation around 4.37 ± 0.07 was measured. This slight increase in sap pH due to the transpiration mechanisms of plants was consistent with the previously reported work in the field. The low-cost measurement system also incorporated a bme680 sensor that allowed the measurement of environmental parameters such as temperature, humidity and VOC concentrations and could send the obtained data wirelessly via an online server.

To simplify the incorporation of this technology within a real-world environment and allow the use by the widest community of farmers, the codes needed for the operation of the final device were translated to XOD, an object-oriented programming tool to allow easy operation by the final user.

Finally, the response of plant tissues towards the implantation of sensing electrodes were studied for the first time. The formation of a lignified film between xylem and sensor surface could be observed. This lignified layer increased the electrical impedance between the reference and working electrodes and led to a significant increase in electrochemical drift.

The results presented in this work represent a step forward for the incorporation of plant biosensors within low resource settings. These sensors have the potential to improve the early diagnosis of plant stress, which could be used for the prevention of abiotic stress and disease spreading. This diagnosis is essential to improve crop yields and tackle the growing demands in food production. The low-cost and easy operation of the devices in this work could speed up the implementation within real-world settings and could be easily modified for the detection of multiple analytes, enabling a multiplexed analysis. Thus, this work shows a plethora of applications in smart farming and Internet of Things (IoT) applied to agriculture.

## Figures and Tables

**Figure 1 biosensors-12-00447-f001:**
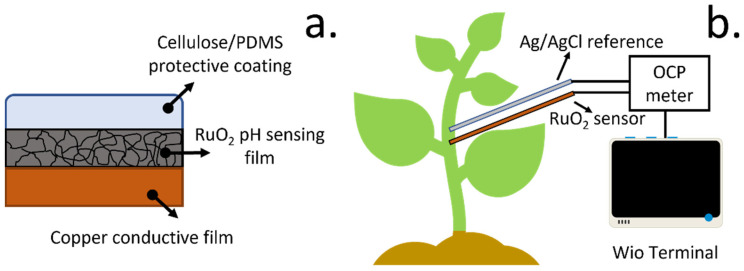
(**a**) Schematic representation of final sensor structure, comprising a copper electrode, where the RuO_x_ films were fabricated by electrodeposition for the sensing of pH, and a cellulose/PDMS coating was applied by aerosol deposition as a protective film, to improve stability. (**b**) After the fabrication of the sensors, the electrodes were implanted inside 4-week tomato plants. Both a solid-state Ag/AgCl reference and RuO_x_ films were incorporated, and the OCP was measured using a Wio Terminal device.

**Figure 2 biosensors-12-00447-f002:**
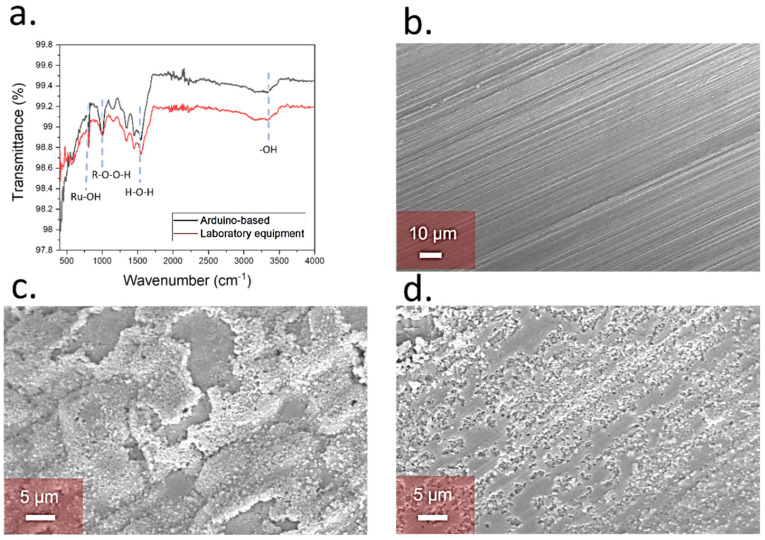
(**a**) FTIR spectrum of the Ruthenium oxide films deposited onto the copper substrates and using laboratory equipment (red), and the atomizer-based device reported in this work (black). The main stretches that indicate the successful deposition of RuO_x_ are highlighted. (**b**) SEM imaging of copper electrodes before the modification through electrodeposition. (**c**) Morphology of the ruthenium oxide-based pH sensing films after electrodeposition using laboratory equipment. (**d**) Results comparison of RuO_x_ deposition onto copper electrodes using an Arduino-based potentiostat.

**Figure 3 biosensors-12-00447-f003:**
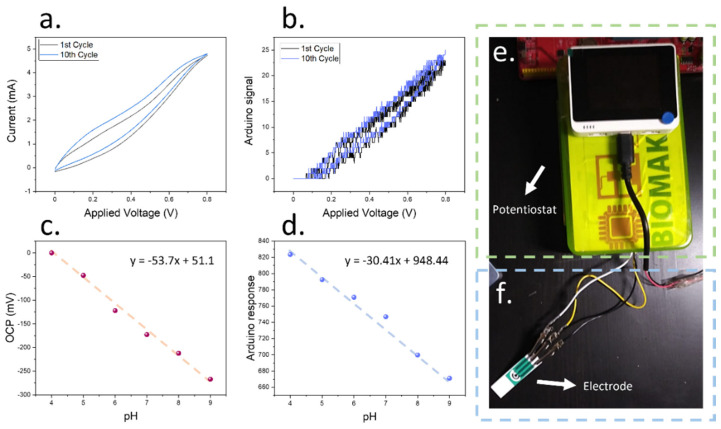
(**a**) Cyclic voltammogram obtained during the electrodeposition of ruthenium oxide films onto copper electrodes using laboratory equipment. The differences between the first and tenth cycles are shown. (**b**) Comparison with cyclic voltammogram obtained using the low-cost potentiostat reported in the present work under the same conditions. (**c**) Calibration plot of the ruthenium oxide sensing films obtained using the low-cost Arduino device reported here using laboratory equipment. The obtained calibration plot is shown. (**d**) Comparison of the calibration results obtained using a RuO_x_ film deposited by a low-cost potentiostat and measuring OCP using an off-the-shelf low-cost device. (**e**) Embodiment of the low-cost potentiostat employing the Wio Terminal device for the fabrication of pH sensing films, including a Wio Terminal device and plastic case where all the electronic components are embedded. (**f**) This potentiostat could be connected to a three-electrode cell incorporating a Ag/AgCl reference, a carbon-based counter and working electrodes for the low-cost deposition of nanomaterials or the sensing.

**Figure 4 biosensors-12-00447-f004:**
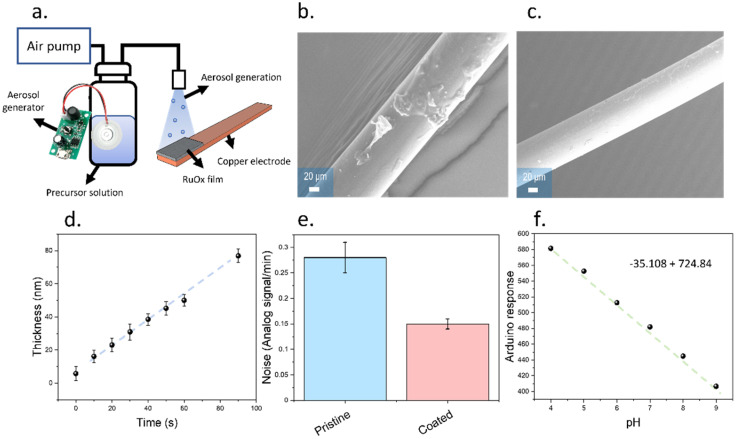
(**a**) Schematic representation of the aerosol system employed within the present work for the deposition of cellulose-based coatings. The ultrasonic atomizer could be used to generate an aerosol which could be directed towards the sample by using an air pump. (**b**) SEM imaging of drop casted films onto copper substrates, evidencing the presence of a high surface roughness. (**c**). The deposition of the cellulose-based coating using the aerosol method led to homogeneous films with controllable thicknesses. This method can also be applied to substrates with different morphologies and the final thickness of the films can be controlled. (**d**) Calibration plot obtained after fabricating cellulose-based films onto the RuO_x_ films using different deposition times. (**e**) Comparison graph of the noise rates obtained using the pristine sensors, only containing the RuO_x_ films, and the cellulose-based electrodes. The error bars indicate the standard deviation from triplicate measurements. (**f**) Calibration plot of the final devices containing RuO_x_ and a cellulose-based coating. The sensitivity achieved increased compared to the one achieved in the case of pristine RuO_x_.

**Figure 5 biosensors-12-00447-f005:**
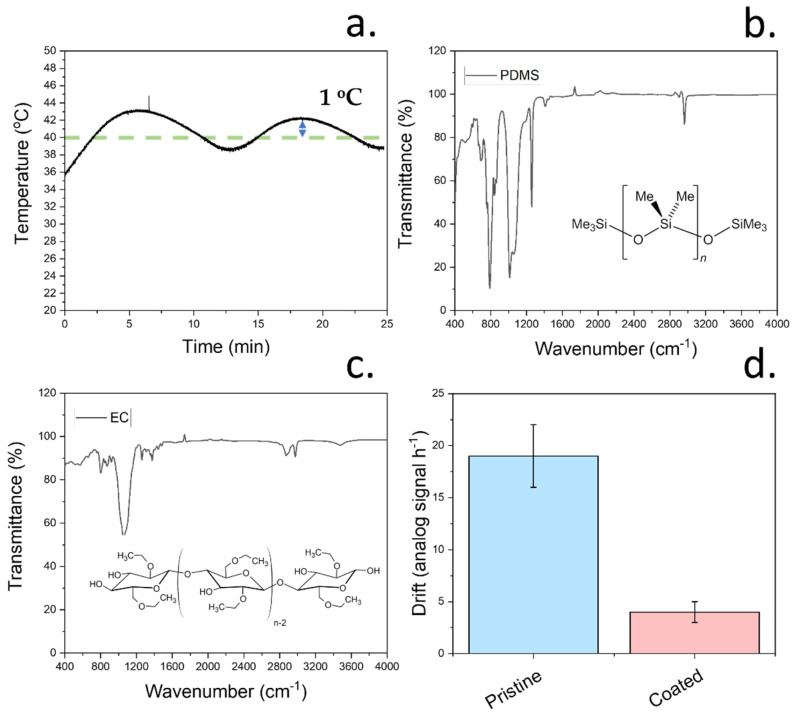
(**a**) Monitoring of substrate temperature using a TM32 sensor with an Arduino device and incorporating a heating element to maintain the surface at a stable temperature. In this case, the substrate temperature was set to 40 °C, and the changes in temperature due to the feedback loop using the temperature sensor can be observed. (**b**) FTIR spectrum of pure PDMS after deposition onto a glass substrate at 40 °C. All the stretches corresponding to PDMS molecules are present. The molecular structure of PDMS is indicated. (**c**) FTIR spectrum of ethyl cellulose after deposition onto a glass substrate at 40 °C. The molecular structure of ethyl cellulose is also indicated. (**d**) Measured drift from pristine and coated pH sensors. The error bars indicate the standard deviation from triplicate measurements.

**Figure 6 biosensors-12-00447-f006:**
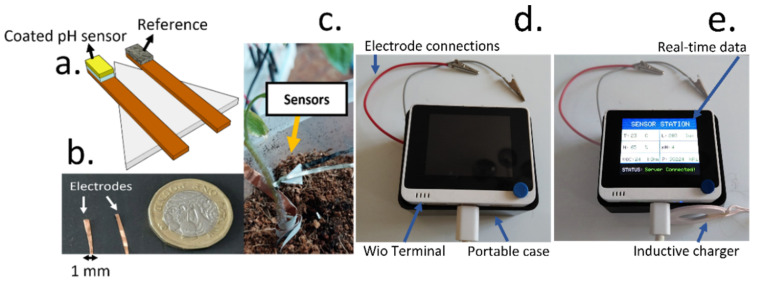
(**a**) Schematic representation of the implantable probe that was inserted inside tomato stems. To improve the implantation, the 1 mm wide electrodes were glued onto a propylene substrate. (**b**) Picture of the pristine copper electrodes prior implantation. A size comparison with a coin is shown, to evidence the low dimensions of the electrodes. Both electrodes were modified with RuO_x_ and Ag/AgCl for the monitoring of pH. (**c**) Final device configuration involving the implanted electrodes inside tomato stems for the monitoring of plant pH. (**d**) Picture of the final device involving a Wio Terminal device, connected to an inductive charger as a power supply, as well as the voltage metre and BME680 environmental sensor (enclosed inside a box). Electrode connections were used to measure the OCP signal from the working and reference electrodes. (**e**) Upon connecting the inductive chargers to the Wio Terminal device, which could also be achieved using a mobile phone, the Wio Terminal can be used without a battery.

**Figure 7 biosensors-12-00447-f007:**
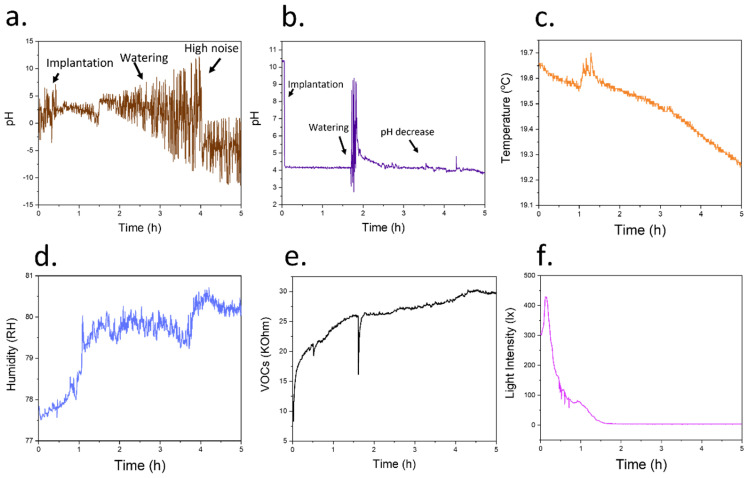
(**a**) Measurement of pH inside the tomato stems using a pristine electrode without the incorporation of a cellulose-based coating. An increasing noise was recorded, with a change upon watering the plants. (**b**) Plot obtained after the recording of pH in a tomato stem using cellulose-based device. The changes in pH upon watering the tomato plant could be determined. In particular, a stable decrease in pH after the watering was obtained. (**c**) The use of a BME680 sensor allowed the determination of environmental temperature. (**d**) The humidity of the devices was additionally measured, evidencing an increase after watering the plant with DI water. This humidity could be used to indicate the water needs from the plant. (**e**) The concentration of VOCs could additionally be assessed through the determination of the resistance of a metal oxide-based gas sensor. In this case, lower values of resistance represent a higher concentration of VOCs. These results indicate a lower concentration of VOC concentrations after watering the plant. (**f**) The Wio Terminal device used within the present work incorporated a light sensor that could be used for the study of day/night cycles in plants. The recorded intensity decreased up to 0 after a few hours.

**Figure 8 biosensors-12-00447-f008:**
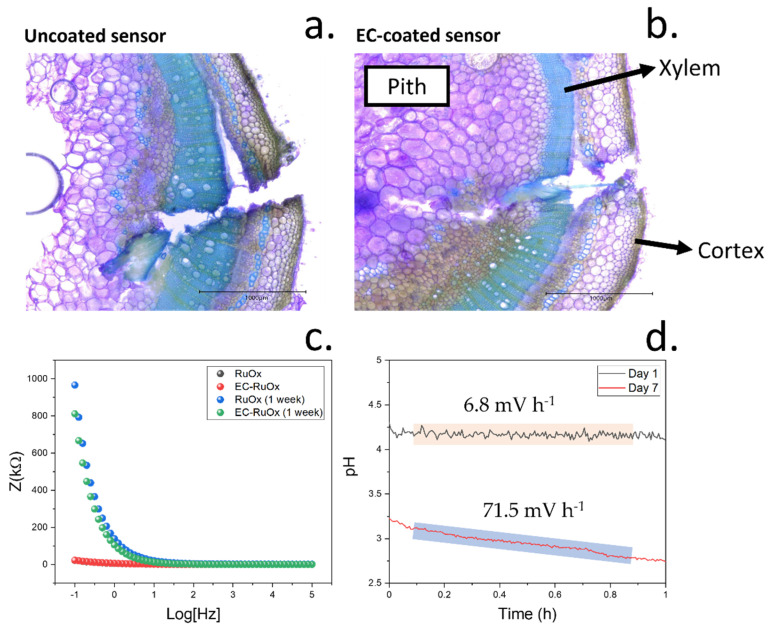
(**a**) Cross-section image of a tomato plant stem where a pristine RuO_x_ electrode had been inserted, evidencing the tissue damage due to the implantation. Three areas from the plant stem are clearly differentiated by the staining process used (cortex, xylem and pith). (**b**) Cross section of tomato plant stem where the cellulose coated electrode had been incorporated. The different plant tissues that can be identified are labelled. (**c**) Differences in electrical impedance obtained from pristine and cellulose-coated electrodes recorded at different frequencies. The impedances on day 1 and day 7 were measured and compared. (**d**) Continuous monitoring of pH in tomatoes on day 1 and day 7, evidencing the increase in signal drift due to the plant healing process.

**Figure 9 biosensors-12-00447-f009:**
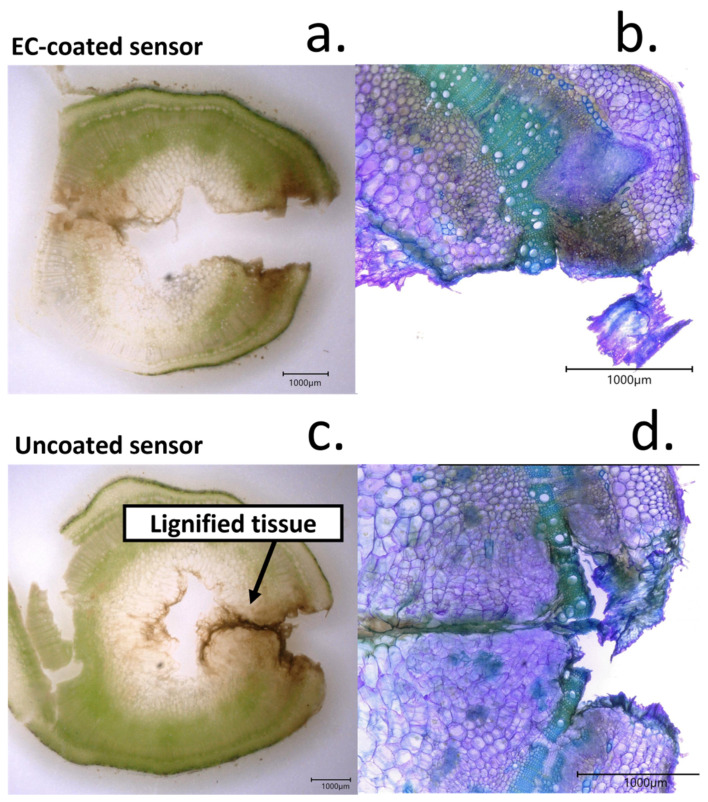
(**a**) Picture of stem cross-section after implantation of pH-sensing electrodes coated with EC/PDMS for 1 week. Brown areas due to the formation of necrotic tissue around the wound area are visible. (**b**) The tissue response from the plant could be imaged by microscopy using Toluidine Blue O stain. This cross-section image was taken from the picture observed in a and evidenced the changes in tissue morphology due to the electrode implantation. (**c**). Picture of tomato stem after the implantation of a cellulose-coated sensor. (**d**) Characterization of tissue responses to exposure of uncoated sensors, only containing RuO_x_ nanoparticles by microscopy.

## Data Availability

Not applicable.

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
