# Peer review of "In Vivo Sensing of pH in Tomato Plants Using a Low-Cost and Open-Source Device for Precision Agriculture†"

_biosensors, 2022, doi:10.3390/bios12070447_

Round 1

Reviewer 1 Report

The authors fabricated a low-cost device for the in vivo detection of pH, humidity, temperature et al in potato plants. The fabrication of the device is quite interesting and the design and characterization of RuO2 nanofilm as the pH sensor is well described. 

I am wondering if the authors can add a few sentences in the introduction and conclusion to describe more clearly the potential applications of their devices. That will make the device more appealing.

Author Response

We would like to thank the reviewer for the positive comments on our manuscript. We hope the manuscript is now suitable for publication. As suggested, we have included a more detailed description of the potential applications of this device within the introduction and the conclusions (P.2 and P.21). For completion, we have also incorporated further details of the potential advantages of using RuOx as the sensing material within the introduction.

Reviewer 2 Report

The manuscript is very interesting and it merits to be published. It is about the design, adaptation, material development, and application for an improved pH sensor in real time and In-Vivo. Please consider the following observations and suggestions before acceptation for publication: i)It should be briefly explained the function of the application of ruthenium oxide-based nanofilms for the in vivo monitoring of pH. In particular in abstract it should be added a phrase related in order to understand the basis. Ii) Please check the redaction and meaning when it was explained the basis of the electrode. In the Introduction it was stated “In the present work, an implanted pH sensor based on ruthenium oxide nanoparticles has been developed”. Revise the sentence because the incorporation of this Nanomaterial is for a targeted property of physical phenomena modification with improved characteristics as for example. It is not clear explained. Rephrase this important point. Iii) In the introduction as well; after mentioning what was presented in the Research work, it was added details about the methodology used. In this part of the manuscript it should not be added these details; however it should be explained the basis of the device function. Iv) Consider to add if it is applicable references for the methods used in the different steps of the chemical modification and sensor design. And if it is completely new these protocols it should be mentioned that the whole was developed in this Research work for the targeted objective. v) A suggestion, please, consider to add a scheme of the sensor with the different compositions and functions. As well, a simple scheme of the incorporation of the sensor in the plant. Schematically for a better understanding of the method and systems developed. It is just a suggestion. Vi) In the section of results it should be added further description of experiments, observations, analysis and discussion in order to conclude and evaluate by this manner highlight the properties obtained. For example,  in section “3.1. Deposition of pH sensing films”. Vii) Consider to reorganize the order of the presentation of results. First show the characterization of the modified substrate. Viii) An additional comment to consider and revise the section of results. There is a nice quantity of experiments developed; however not so much description and discussion of each graph showed as for example. If you check per graph there is a brief sentence of each figure.  

Author Response

We are delighted to hear that the reviewer found this manuscript very interesting. We have addressed the suggested reviews accordingly, and we hope that the manuscript is now suitable for publication.

1.- should be briefly explained the function of the application of ruthenium oxide-based nanofilms for the in vivo monitoring of pH. In particular in abstract it should be added a phrase related in order to understand the basis.

An in-depth explanation of why ruthenium was chosen as the sensing material, and the desirable properties that it shows compared to other compounds has been provided in the introduction (P.2, L26). These properties have been highlighted in the abstract as suggested (P.1., L16).

2.- Please check the redaction and meaning when it was explained the basis of the electrode. In the Introduction it was stated “In the present work, an implanted pH sensor based on ruthenium oxide nanoparticles has been developed”. Revise the sentence because the incorporation of this Nanomaterial is for a targeted property of physical phenomena modification with improved characteristics as for example. It is not clear explained. Rephrase this important point.

We would like to thank the reviewer for flagging this. The sentence has been rephrased as suggested by the reviewer “In the present work, a pH sensor based on electrodeposited ruthenium oxide na-noparticles onto copper electrodes and containing a protective cellulose:PDMS film has been fabricated.” (P.2. L48.).

3.- In the introduction as well; after mentioning what was presented in the Research work, it was added details about the methodology used. In this part of the manuscript it should not be added these details; however it should be explained the basis of the device function.

The last paragraph from the introduction has been revised as suggested. Details from the methodology has been removed, and the work has been explained in terms of the device function (P.3.L.2.).

4.- Consider to add if it is applicable references for the methods used in the different steps of the chemical modification and sensor design. And if it is completely new these protocols it should be mentioned that the whole was developed in this Research work for the targeted objective.

References from the chemical modification and sensor device have been added to the text within the methodology. We have also included details on the staining method for our biological work where relevant.

5.- v) A suggestion, please, consider to add a scheme of the sensor with the different compositions and functions. As well, a simple scheme of the incorporation of the sensor in the plant. Schematically for a better understanding of the method and systems developed. It is just a suggestion.

We agree with the reviewer suggestion. We have added a scheme of the different compositions and functions of the sensor layers (Figure 1.a., P.5., L.1.). A simple scheme of the incorporation in the plant has also been added, where ethe implantation of both the working and reference electrodes is shown (Figure 1.b. P.5., L.1.).

6.- In the section of results it should be added further description of experiments, observations, analysis and discussion in order to conclude and evaluate by this manner highlight the properties obtained. For example,  in section “3.1. Deposition of pH sensing films”.

As suggested by the reviewer, we have added a more detailed explanation of the experiments, including a more detailed description of the results in the results and discussion section. For completion, we have also added a more extensive analysis of the impedance data recorded in vivo, including calculations of electrical and ionic resistance, as indicated by reviewer 3.

7.- Consider to reorganize the order of the presentation of results. First show the characterization of the modified substrate.

We agree with the reviewer suggestion of re-organising the sections to improve the presentation of results. We have moved the materials characterisation section, including the result from SEM and FTIR first, followed by the electrochemical characterisation of the deposited films.

8.- An additional comment to consider and revise the section of results. There is a nice quantity of experiments developed; however not so much description and discussion of each graph showed as for example. If you check per graph there is a brief sentence of each figure.  

We would like to thank the reviewer for this comment on the graphs. We have now added a more in-depth description of the results from the graphs in the results section. Especially on section 3.4., where the results from the environmental monitoring are shown.

Reviewer 3 Report

The authors developed a RuOx-based sensor for xylem sap pH sensing. Most importantly, they applied a cellulose/PDMS coating to achieve more reliable in-vivo signal recording with lower OCP drift and less plant tissue damage. Overall, it is a novel and comparably thorough study with a broad impact.  However, the reviewer has some minor concerns of this manuscript before its publications. 

1). It is very important to show a reproducible sensing performance improvement when using cellulose/PDMS coating. So please include error bar in the in-vitro characterization data of fig. 3e and fig. 4d. 

2). In the related context to explain the EIS in fig. 7c, it doesn't make sense to compare average impedance value since impedance levels at low and high f regions have different electrochemical meaning. Please discuss the results separately in terms of electron-transfer dominated (low f) and electron-transport dominated (high-f) situations.

3). In fig.7d, it indicates a ~70mV/h drift after 7 days-implementation. However, the pH only decreases 1 (~50 mV) over 7 days despite rapid drifting. Please explain it. 

Author Response

We are delighted to see that the reviewer found our work novel and thorough. We have work to address the proposed reviews, and we hope that the manuscript is now suitable for publication.

1.- It is very important to show a reproducible sensing performance improvement when using cellulose/PDMS coating. So please include error bar in the in-vitro characterization data of fig. 3e and fig. 4d.

We agree with the reviewer that the reproducibility of the sensing performance is key. As suggested, be have now included error bars for the in vitro characterisation data (Figure 4., P.10., L.1.) and Figure 5., P.12., L.1.).

2.- In the related context to explain the EIS in fig. 7c, it doesn't make sense to compare average impedance value since impedance levels at low and high f regions have different electrochemical meaning. Please discuss the results separately in terms of electron-transfer dominated (low f) and electron-transport dominated (high-f) situations.

We agree with the reviewer that a more detailed explanation of the impedance plots would benefit the discussion of our biological results. A comparison of impedance plots have been added considering the results from high and low frequency ranges separately. For completion of this study, we have estimated the changes in intracellular and extracellular resistance between the electrodes by simulating the results using a reported equivalent circuit. These new results have been discussed in the manuscript (P.16., L.21., and P.16., L.43.).

3). In fig.7d, it indicates a ~70mV/h drift after 7 days-implementation. However, the pH only decreases 1 (~50 mV) over 7 days despite rapid drifting. Please explain it.

We would like to thank the reviewer for flagging this result. The drift measurements were taken on day 1, and day 7 of the in vivo experiments. However, the OCP values were not being recorded continuously throughout this period, only during the time shown in the graphs. As such, we expect the drift obtained on day 7 to be the result of the initial drift from initializing the OCP sensing device, and the wound healing mechanism of the plant, which restricts the sap flow onto the sensing area. In addition, it is expected that the wound healing mechanism in the plant involves the production of metabolites and the release of electrolytes which might compromise the composition of the sensing films within the sensing area. These phenomena have been discussed on the manuscript (P.19., L.22.).